# ACLP Activates Cancer-Associated Fibroblasts and Inhibits CD8+ T-Cell Infiltration in Oral Squamous Cell Carcinoma

**DOI:** 10.3390/cancers15174303

**Published:** 2023-08-28

**Authors:** Shohei Sekiguchi, Akira Yorozu, Fumika Okazaki, Takeshi Niinuma, Akira Takasawa, Eiichiro Yamamoto, Hiroshi Kitajima, Toshiyuki Kubo, Yui Hatanaka, Koyo Nishiyama, Kazuhiro Ogi, Hironari Dehari, Atsushi Kondo, Makoto Kurose, Kazufumi Obata, Akito Kakiuchi, Masahiro Kai, Yoshihiko Hirohashi, Toshihiko Torigoe, Takashi Kojima, Makoto Osanai, Kenichi Takano, Akihiro Miyazaki, Hiromu Suzuki

**Affiliations:** 1Department of Molecular Biology, Sapporo Medical University School of Medicine, Sapporo 060-8556, Japankubo-t@grape.plala.or.jp (T.K.); kai@sapmed.ac.jp (M.K.); 2Department of Oral Surgery, Sapporo Medical University School of Medicine, Sapporo 060-8543, Japan; 3Department of Otolaryngology-Head and Neck Surgery, Sapporo Medical University School of Medicine, Sapporo 060-8543, Japan; 4Department of Pathology, Sapporo Medical University School of Medicine, Sapporo 060-8556, Japantorigoe@sapmed.ac.jp (T.T.);; 5Department of Head and Neck Oncology, Sapporo Teishinkai Hospital, Sapporo 065-0033, Japan; 6Department of Cell Science, Research Institute of Frontier Medicine, Sapporo Medical University School of Medicine, Sapporo 060-8556, Japan; ktakashi@sapmed.ac.jp

**Keywords:** oral cancer, tumor microenvironment, cancer-associated fibroblast, collagen, tumor-infiltrating lymphocyte

## Abstract

**Simple Summary:**

Cancer-associated fibroblasts (CAFs) are a major component of the stroma in oral squamous cell carcinoma (OSCC) and are considered important therapeutic targets. In this study, we demonstrated that aortic carboxypeptidase-like protein (ACLP) is highly expressed in CAFs of OSCC, thereby activating them. Cancer cells induce ACLP expression in CAFs through the TGF-β1 signaling pathway, and CAF-derived ACLP enhances the migration and infiltration of cancer cells. Furthermore, ACLP co-expresses with collagen and shows an inverse correlation with tumor infiltration of CD8+ T lymphocytes. Our data suggest that targeting ACLP could be a potential approach for stromal-targeted therapy and a novel target for cancer immunotherapy.

**Abstract:**

We previously showed that upregulation of adipocyte enhancer-binding protein 1 (*AEBP1*) in vascular endothelial cells promotes tumor angiogenesis. In the present study, we aimed to clarify the role of stromal *AEBP1*/ACLP expression in oral squamous cell carcinoma (OSCC). Immunohistochemical analysis showed that ACLP is abundantly expressed in cancer-associated fibroblasts (CAFs) in primary OSCC tissues and that upregulated expression of ACLP is associated with disease progression. Analysis using CAFs obtained from surgically resected OSCCs showed that the expression of *AEBP1*/ACLP in CAFs is upregulated by co-culture with OSCC cells or treatment with TGF-β1, suggesting cancer-cell-derived TGF-β1 induces *AEBP1*/ACLP in CAFs. Collagen gel contraction assays showed that ACLP contributes to the activation of CAFs. In addition, CAF-derived ACLP promotes migration, invasion, and in vivo tumor formation by OSCC cells. Notably, tumor stromal ACLP expression correlated positively with collagen expression and correlated inversely with CD8+ T cell infiltration into primary OSCC tumors. Boyden chamber assays suggested that ACLP in CAFs may attenuate CD8+ T cell migration. Our results suggest that stromal ACLP contributes to the development of OSCCs, and that ACLP is a potential therapeutic target.

## 1. Introduction

There were 377,713 new cases of oral cancer worldwide in 2020 (2.0% of all cancers) and 177,757 deaths caused by this disease (1.8% of all cancers) [1]. About 90% of oral cancers are histologically classified as squamous cell carcinoma [2]. Cancers of the tongue are the most common, accounting for 30–40% of oral cancers [2]. A multidisciplinary approach including surgery, chemotherapy, and radiation therapy is recommended for the treatment of oral cancers. But, while these treatments are effective for early-stage cancers, they are sometimes insufficient for advanced or recurrent disease. Hence, the five-year survival rate for patients with oral cancer is only about 60% [3,4]. In recent years, targeted molecular therapy and immunotherapy have been applied to oral cancer. For instance, cetuximab, a monoclonal antibody targeting epidermal growth factor receptor (EGFR), was approved as a targeted drug for unresectable or chemoresistant oral cancers [5]. More recently, pembrolizumab, an anti-PD-1 antibody, was also approved as an immune checkpoint inhibitor with the same indication [6]. Despite these recent advances, the success rate in treating oral cancer remains unsatisfactory, which highlights the need to identify new therapeutic targets [7,8].

Cancer consists of the tumor microenvironment (TME) composed of cancer cells, stromal cells, and the extracellular matrix (ECM) that surrounds them [9]. Cancer-associated fibroblasts (CAFs) are the major cells that constitute the TME, and their presence is known to promote tumorigenesis in many tumor types [10]. Moreover, CAFs interact with tumor-infiltrating immune cells and other immune components within the TME by secreting various cytokines, growth factors, chemokines, and exosomes to make the TME immunosuppressive and to allow cancer cells to escape immune surveillance [11,12,13]. These findings make CAFs a promising potential therapeutic target. However, the role of CAFs and their usefulness as a therapeutic target in oral cancer have not been clearly established.

Adipocyte enhancer-binding protein 1 (AEBP1) has been initially identified as a transcriptional repressor involved in adipocyte differentiation [14]. One of the isoforms of AEBP1, also known as aortic carboxypeptidase-like protein (ACLP), is a secreted ECM protein that associates with collagen and is reportedly involved in wound healing and fibrosis [15,16,17,18]. Recent studies have found that AEBP1/ACLP also plays an important role in various malignancies. For instance, the expression of *AEBP1* is frequently upregulated in primary glioblastoma, and silencing of *AEBP1* induces apoptosis in glioblastoma cell lines [19]. In melanoma cells, upregulation of AEBP1 confers resistance to BRAF inhibition [20]. AEBP1 also reportedly promotes colon and gastric cancer cell proliferation and metastasis by activating the NF-κB pathway [21,22]. We previously reported that *AEBP1* is highly expressed in endothelial and stroma cells in colorectal cancer (CRC) and that upregulated expression of *AEBP1* in endothelial cells facilitates tumor angiogenesis [23]. We also reported that *AEBP1*/ACLP is a marker of CAFs in CRC [24]. In the present study, we aimed to elucidate the function of *AEBP1*/ACLP and its usefulness as a therapeutic target in oral squamous cell carcinoma (OSCC).

## 2. Materials and Methods

### 2.1. Tissue Samples and Cell Culture

Tissues of primary tongue OSCC were collected from Japanese patients who were treated at the Departments of Otolaryngology—Head and Neck Surgery (*n* = 49) and Oral Surgery (*n* = 49) in Sapporo Medical University Hospital between April 2009 and April 2019 (Appendix A). Informed consent was obtained from all patients before collection of the specimens. TNM stages were evaluated according to the eighth edition of the American Joint Committee on Cancer (AJCC) staging system. OSCC cell lines were obtained from the Japanese Collection of Research Bioresources (Tokyo, Japan) and cultured as described previously [25,26]. CAFs (CAF1, CAF2, and CAF3) were isolated from 3 OSCC patients. To obtain CAFs, surgically excised OSCC tissues were minced using sterilized scalpels. Scratches were made on the bottom of a 10 cm dish using a pipette tip, and the minced tissue fragments were placed on it, followed by cultivation in Dulbecco’s modified Eagle’s medium (DMEM) supplemented with 20% fetal bovine serum (FBS) and 1% antibiotic–antimycotic (Thermo Fisher Scientific, Waltham, MA, USA) or in Fibroblast Growth Medium 2 (FGM2; PromoCell, Heidelberg, Germany) with a Fibroblast Growth Medium 2 Supplement Pack (PromoCell) and 1% antibiotic–antimycotic (Thermo Fisher Scientific). For co-culture experiments, CAFs (1 × 10^5^ cells in 6-well plates) were indirectly co-cultured with OSCC cells (1.2 × 10^6^ cells in culture inserts) for 96 h. Where indicated, CAFs were treated for 48 h with 10 ng/mL transforming growth factor-β1 (TGF-β1; PeproTech, Cranbury, NJ, USA). CD8+ T cells were obtained from a healthy donor, as described [27]. Briefly, peripheral blood mononuclear cells (PBMCs) were isolated from a healthy donor using Lymphoprep (Cosmo Bio, Tokyo, Japan) according to the manufacturer’s instructions and cultured in AIM-V serum-free medium (Thermo Fisher Scientific) containing 10% human AB serum (Biowest, Nuaillé, France). CD8+ T cells were isolated from PBMCs using the MACS separation system (Miltenyi Biotech, Bergisch Gladbach, Germany) with anti CD8 mAb coupled with magnetic microbeads according to the manufacturer’s instructions. CD8+ T cells were activated by culturing them in AIM-V medium containing 1 μg/mL of phytohemagglutinin P (PHA-P; Wako, Osaka, Japan) and 100 U/mL of human recombinant IL-2 (R&D Systems, Minneapolis, MN, USA) for 3 days. Informed consent was obtained from the healthy donor before collection of the sample. This study was approved by the Institutional Review Board at Sapporo Medical University (No. 322–38).

### 2.2. Immunohistochemistry

Immunohistochemical staining was carried out as described previously [28]. A mouse anti-Human ACLP/AEBP1 mAb (1:100 dilution, LS-C133036; LSBio), mouse anti-α-smooth muscle antigen (α-SMA) mAb (1:50 dilution, M0851; CiteAb, Bath, UK), and mouse anti-CD8 mAb (clone C8/144B; DAKO) were used. Areas positive for ACLP and α-SMA were measured using ImageJ software ver. 1.52 (NIH, Bethesda, MD, USA) as described [29]. Collagen I was stained using a Picrosirius Red Stain Kit (Polysciences, Warrington, PA, USA) according to the manufacturer’s instructions.

### 2.3. Reverse-Transcription PCR

Total RNA was extracted using an RNeasy Mini kit (Qiagen, Hilden, Germany). Single-stranded cDNA was prepared using a PrimeScript RT Reagent Kit with gDNA Eraser Perfect Real Time (Takara Bio Inc., Kusatsu, Japan). Reverse-transcription PCR (RT-PCR) was performed as described previously [30]. Quantitative RT-PCR (qRT-PCR) was performed using PowerUp SYBR Green Master Mix (Thermo Fisher Scientific) with a QuantStudio 3 Real-Time PCR System (Thermo Fisher Scientific). β-actin (ACTB) was used as an endogenous control. Primer sequences are listed in Appendix A.

### 2.4. Western Blot Analysis

Total cell lysate extraction and Western blot analysis were carried out as described previously [31]. A rabbit anti-AEBP1 mAb (1:3000 dilution, ab168355; Abcam, Cambridge, UK) and mouse anti-β-actin mAb (1:3000 dilution, clone AC-15; Sigma-Aldrich, Darmstadt, Germany) were used.

### 2.5. Tumor-Conditioned Medium

To prepare tumor-conditioned medium (TCM), OSCC cells were cultured in growth medium supplemented with 10% FBS, after which TCM was prepared as described previously [23]. CAFs (1.0 × 10^5^ cells in 6-well plates) were treated for 24 h with TCM.

### 2.6. siRNA and Expression Vector

For knockdown of *AEBP1*, 1 × 10^6^ cells were transfected with Silencer Select Pre-designed siRNA (100 pmol each; AEBP1 siRNA1, s1145; AEBP1 siRNA2, s1146; Thermo Fisher Scientific) or Silencer Select Negative Control No. 1 siRNA (Thermo Fisher Scientific) using a TransIT-X2 Dynamic Delivery System (Mirus Bio, Madison, WI, USA). A lentiviral vector encoding full-length *AEBP1* was constructed as described previously [23].

### 2.7. Collagen Gel Contraction Assays

CAFs were transfected with siRNAs as described above, after which gel contraction assays were performed. Mixtures containing 3 × 10^5^ CAFs and 1 mL of type I collagen gel (Cellmatrix type I-A; Nitta Gelatin, Osaka, Japan) were placed in 12-well plates. The collagen gel mixture was incubated at 37 °C for 30 min to polymerize the gel, after which 1 mL of serum-free medium was added. After incubation for 48 h at 37 °C, the surface area of the gel was measured. The contraction rate was calculated using the formula (1—gel surface area/well surface area) × 100%.

### 2.8. Gene Expression Microarray Analysis

CAFs were transfected with siRNAs as described above, and total RNA was extracted 72 h after transfection. Gene expression microarray analysis was then performed using a SurePrint G3 Human GE 8 × 60 K v2 microarray (Agilent Technologies, Santa Clara, CA, USA) as described previously [23]. Data were analyzed using GeneSpring GX version 13 (Agilent Technologies) and Gene set enrichment analysis (GSEA; Broad Institute, Boston, MA, USA). The Gene Expression Omnibus accession number for the microarray data is GSE234220.

### 2.9. Transwell Migration and Invasion Assays

Transwell migration and invasion assays were performed as described previously [29]. To assess the effect of CAF-derived ACLP on OSCC cell migration, CAFs were transfected with siRNAs as described above. OSCC cells (5 × 10^4^ cells) in serum-free medium were then seeded into the upper chamber and CAFs (2 × 10^4^ cells) in culture medium with 20% FBS were added to the lower well. To assess the effect of recombinant ACLP on OSCC cell migration and invasion, 5 × 10^4^ OSCC cells in serum-free medium with or without recombinant human ACLP (10 ng/mL; R&D Systems) were seeded into the upper chamber, and culture medium with 10% FBS was added to the lower well. After incubation for 24 h at 37 °C, migrating or invading cells on the lower surface of the filter were fixed and stained using a Diff-Quik staining kit (Sysmex, Tokyo, Japan).

### 2.10. Three-Dimensional Culture

CAFs were transfected with siRNAs or infected with lentiviral vectors as described above and incubated for 24 h. Three-dimensional (3D) culture was performed as described [32]. Samples consisting of a mixture of 2.5 × 10^5^ CAFs and 1 mL of type I collagen gel (Nitta Gelatin) were placed in 12-well plates and incubated for 30 min. OSCC cells (1 × 10^6^ cells) were seeded onto the gel and cultured in DMEM supplemented with 20% FBS, after which the gels were transferred to 6-well plates. After 6 days, the gels were fixed with 10% formaldehyde and stained with hematoxylin and eosin. Invasion areas were assessed using ImageJ software ver. 1.52 (NIH) in five randomly selected fields per gel.

### 2.11. Cell Viability Assays

CAFs were transfected with siRNAs as described above and seeded into 96-well plates (1 × 10^4^ cells per well). To assess the effect of ACLP on chemosensitivity, OSCC cells in culture medium with 10% FBS and recombinant human ACLP (0 to 10 ng/mL; R&D Systems) were seeded into 96-well plates (1 × 10^4^ cells per well) and incubated for 48 h, after which they were treated with cisplatin (CDDP; 0 to 4.0 μg/mL) for an additional 48 h. Cell viability was then assessed using a Cell Counting kit-8 (Dojindo, Kumamoto, Japan) according to the manufacturer’s instructions.

### 2.12. Xenograft Study

CAFs were infected with lentiviral vectors as described above. To analyze the effect of co-transplantation of OSCC cells and CAFs on xenograft formation, 1 × 10^5^ OSCC cells and 2 × 10^5^ CAFs were suspended in PBS plus 0.2 mL of Matrigel (Corning Inc., Corning, NY, USA) and injected subcutaneously into the dorsal flank of 4-week-old female BALB/cAJcl-nu mice. Tumor size was measured every 3 days using digital calipers, and tumor volume was calculated using the formula length × width^2^/2. Mice were sacrificed and tumors were harvested 14 days after transplantation. All animal experiments were conducted in compliance with a protocol approved by the Institutional Animal Care and Use Committee of Sapporo Medical University (No. 19-062_22-027_22-068).

### 2.13. CD8+ T Cell Migration Assays

CAFs (3.0 × 10^3^ cells) in 50 µL of FGM2 were seeded into the upper chamber of a Transwell plate (Corning HTS Transwell 96 wells, #CLS3388, Corning Inc.), and 200 µL of FGM2 was added to the lower well. After 24 h, CAFs were transfected with siRNAs as described above and incubated for 48 h at 37 °C. Medium in the upper chamber was subsequently replaced with 50 µL of AIM V serum-free medium (Thermo Fisher Scientific) with 10% human serum (Biowest, Nuaillé, France) containing 1 × 10^5^ CD8+ T lymphocytes. The medium in the lower well was replaced with 200 µL of culture medium with 10% human serum and recombinant human CXCL10 (1 ng/mL, Shenandoah Biotechnology, Warminster, PA, USA). After incubation for an additional 24 h, cells in the lower chamber were counted using a Countess C10281 automated cell counter (Thermo Fisher Scientific).

### 2.14. Statistical Analysis

Fisher’s exact test was performed for analysis of categorical data. Student’s t-tests or ANOVA with post hoc tests were used to analyze quantitative variables. The Kaplan–Meier method was used for survival analysis. A log-rank test for 2-group comparisons was used to compare survival curves. Values of *p* less than 0.05 (2-sided) were considered statistically significant. Statistical analyses were performed using GraphPad Prism 5 (GraphPad Software, La Jolla, CA, USA).

## 3. Results

### 3.1. Elevated Expression of ACLP in Stromal Cells from Primary Tumors

To evaluate the function of ACLP in the tumor microenvironment of OSCC, we first analyzed the expression of ACLP and the CAF marker α-SMA immunohistochemically in a series of 49 primary tongue squamous cell carcinoma tissues. We found that the expression of ACLP was higher in stromal cells within the tumor tissues than in adjacent normal tissues (Figure 1A,B). Notably, we detected co-expression of ACLP and α-SMA in the tumor stromal cells, suggesting CAFs are the major source of ACLP expression in OSCC tissues (Figure 1A). Quantitative analysis revealed a significant positive correlation between levels of ACLP expression and those of α-SMA expression within the tumor tissues (Figure 1C). Moreover, higher levels of ACLP expression correlated with advanced clinicopathological characteristics, whereas they were not associated with overall patient survival (Figure 1D,E, Appendix A). We also analyzed ACLP expression in another independent cohort of tongue squamous cell carcinoma tissues (*n* = 49) and found that, again, elevated ACLP expression correlated with advanced characteristics (Appendix A). Analysis using RNA-seq data of The Cancer Genome Atlas (TCGA) revealed significant positive associations between *AEBP1* expression levels and levels of three representative CAF markers, actin alpha 2 (*ACTA2*), fibroblast activation protein (*FAP*), and platelet-derived growth factor receptor beta (*PDGFRB*), in primary HNSCCs (Appendix A). In addition, gene ontology and KEGG pathway analyses using the RNA-seq datasets suggested that genes co-expressed with *AEBP1* are associated with the collagen metabolic process, extracellular structure organization, integrin-mediated signaling pathway, and ECM receptor interaction (Appendix A). These results suggest that elevated expression of ACLP is associated with the progression of OSCCs, and that *AEBP1*/ACLP may play an important role in the tumor microenvironment.

### 3.2. Induction of AEBP1/ACLP in CAFs by OSCC Cells

ACLP is encoded by the longer variant of the *AEBP1* gene (*AEBP1* variant 1, Figure 2A). We previously reported that vascular endothelial cells dominantly express variant 1, while CRC cells show expression of variant 2 [23]. RT-PCR analysis using a primer pair, which is able to discriminate between *AEBP1* variants, revealed that normal fibroblasts and CAFs express variant 1, while the majority of OSCC cell lines express variant 2 (Figure 2B). Our RT-PCR analysis also suggested that levels of AEBP1 expression in fibroblasts and CAFs were significantly higher than those in cancer cells (Figure 2B). In the previous study, we found that *AEBP1* expression in vascular endothelial cells was upregulated by co-culture with cancer cells or treatment with TGF-β1 [23]. We therefore cultured CAFs with or without OSCC cells and observed elevated levels of *AEBP1* mRNA in the co-cultured CAFs (Figure 2C). We also treated CAFs with TCM derived from a series of OSCC cell lines and found that TCM from multiple OSCC cell lines upregulated *AEBP1* in CAFs (Figure 2D). Likewise, treating CAFs with TGF-β1 significantly upregulated levels of both *AEBP1* mRNA and ACLP protein (Figure 2E,F, Appendix A). The larger bands in the Western blot represent a glycosylated form of the ACLP protein.

### 3.3. Functional Analysis of ACLP in CAFs

To clarify the function of ACLP in CAFs, we transfected the cells with siRNAs targeting *AEBP1*, thereby depleting both *AEBP1* mRNA and ACLP protein (Figure 3A,B, Appendix A). We found that *AEBP1*/ACLP knockdown significantly suppressed collagen gel contraction by CAFs, which suggests ACLP contributes to ECM remodeling by CAFs (Figure 3C). We also found that *AEBP1*/ACLP knockdown inhibited CAF proliferation (Figure 3D). GSEA using microarray data obtained from CAFs with or without *AEBP1*/ACLP knockdown revealed that depletion of *AEBP1*/ACLP significantly downregulated E2F targets, G2/M checkpoint genes, and MYC target genes, suggesting *AEBP1*/ACLP may promote cell cycle progression in CAFs (Figure 3E). Gene ontology analysis also suggested that cell-cycle-related genes were enriched among downregulated genes (Figure 3F). We also observed a tendency for genes associated with the p53 pathway to be upregulated by *AEBP1*/ACLP depletion in CAFs (Figure 3E).

### 3.4. ACLP Promotes Cancer Cell Migration and Invasion and In Vivo Tumor Formation

To investigate the effects of CAF-derived ACLP on cancer cells, we first performed Boyden chamber assays, which revealed that *AEBP1*/ACLP knockdown in CAFs attenuated OSCC cell migration (Figure 4A). Similarly, collagen gel invasion assays showed that *AEBP1*/ACLP knockdown in CAFs suppressed invasion by OSCC cells (Figure 4B), while ectopic expression of *AEBP1*/ACLP in CAFs had the opposite effect (Figure 4C,D, Appendix A). These results suggest that ACLP secreted by CAFs activates migration and invasion by OSCC cells. Consistent with that idea, treating OSCC cells with recombinant ACLP upregulated their migration and invasiveness (Figure 4E,F, Appendix A). We also found that treatment with recombinant ACLP upregulated VIM, TWIST, and CDH2 while suppressing CDH1 in OSCC cells. This suggests ACLP may induce epithelial mesenchymal transition (EMT, Figure 4G). As a number of studies have shown that CAFs contribute to chemoresistance, we also tested whether ACLP is involved in OSCC cell resistance to CDDP. We found that recombinant ACLP moderately increased the viability of OSCC cells treated with CDDP, suggesting ACLP may confer CDDP resistance in OSCC cells (Figure 4H).

We then used a xenograft model to evaluate the effect of ACLP on in vivo tumor formation by OSCC cells. We co-transplanted nude mice with OSCC cells and CAFs with or without ectopic expression of *AEBP1*/ACLP and found that CAF-derived ACLP promoted xenograft formation (Figure 5A,B). Immunohistochemical analysis revealed higher levels of stromal α-SMA expression in tumors with ACLP overexpression than in control tumors, which suggests ACLP contributes to the expansion of the tumor stroma (Figure 5C).

### 3.5. Expression of ACLP Inversely Correlates with Intratumoral Filtration of CD8+ T Lymphocytes

As mentioned above, *AEBP1* is co-expressed with genes involved in collagen metabolic processes in primary HNSCC tissues (Appendix A). Further analysis using TCGA datasets revealed that levels of *AEBP1* expression correlated strikingly with those of collagen family genes, including *COL1A1*, *COL1A2*, *COL3A1*, *COL6A1*, and *COL6A2* (Figure 6A, Appendix A). Immunohistochemical analysis also confirmed the co-expression of ACLP and collagen I in stromal cells in primary OSCC tissues and xenograft tumors (Figure 6B, Appendix A). Notably, we found that ACLP expression correlates inversely with intratumoral infiltration of CD8+ T cells (Figure 6C,D, Appendix A). This suggests elevated ACLP expression inhibits intratumoral infiltration by cytotoxic lymphocytes. To test this possibility, we performed Boyden chamber assays to assess whether CAFs placed in the bottom of the upper chamber affect the migration of CD8+ T cells (Figure 6E). We found that *AEBP1*/ACLP knockdown in CAFs significantly upregulated lymphocyte migration (Figure 6E).

## 4. Discussion

In the present study, we show that AEBP1/ACLP is abundantly expressed in CAFs in primary OSCCs and that CAF-derived AEBP1/ACLP contributes to the disease progression. AEBP1 was first identified as a transcriptional repressor with carboxypeptidase activity involved in adipogenesis [14,33]. Since then, it has also been associated with various other biological processes, including cholesterol homeostasis, inflammation, and obesity [34,35,36]. ACLP is a non-nuclear AEBP1 isoform with an N terminal extension and is upregulated during vascular smooth muscle differentiation [37]. ACLP is abundantly expressed in ECM, and ACLP knockout mice show impaired abdominal wall development and deficient wound healing, which suggests ACLP is essential for embryonic development and dermal wound healing [16]. ACLP is expressed at high levels in fibrotic lung tissues and is a potential therapeutic target for the treatment of pulmonary fibrosis [17]. Notably, ACLP promotes lung fibroblast-to-myofibroblast differentiation and collagen expression [38]. Taken together with these observations, our results suggest that elevated ACLP expression may drive oral tumorigenesis via CAF activation. Consistent with that idea, Li et al. recently demonstrated that ACLP activates CAFs and promotes metastasis in pancreatic cancer, further supporting our hypothesis [39]. Our data also suggest that ACLP may promote the proliferation of CAFs through modulating the cell cycle. However, further study is necessary to clarify the molecular function of ACLP in OSCC.

Our qRT-PCR results suggest that, of the two transcriptional variants of AEBP1, variant 1 (encoding ACLP) is dominantly expressed in CAFs. By contrast, the majority of OSCC cell lines dominantly express variant 2 (encoding AEBP1) but do not express ACLP. Similarly, we have shown that CRC cells preferentially express AEBP1 and do not express ACLP [23]. As described above, recent studies have shown that AEBP1 plays an oncogenic role in cancer cells of various origins, including glioma, melanoma, gastric cancer, and CRC cells [19,20,21,22]. In addition, other recent studies have shown that miR-214 inhibits AEBP1 expression and increases the chemosensitivity in CRC cells, and that AEBP1 knockdown induces ferroptosis in cisplatin-resistant oral cancer cells [40,41]. These results suggest that both AEBP1 and ACLP contribute to the development and progression of various malignancies, although their molecular functions and the cell types in which they are expressed may differ.

We observed that high ACLP expression correlates inversely with intratumoral infiltration of CD8+ T cells. We also noted that levels of AEBP1/ACLP expression correlate significantly with those of collagen in primary tumors. Multiple studies have shown that collagen may inhibit intratumoral infiltration of lymphocytes. For instance, levels of stromal collagen correlate inversely with infiltration by CD8+ T cells in gastric cancer [42]. Similarly, fibrosis is negatively associated with tumor-infiltrating lymphocytes in triple negative breast cancer [43,44]. This suggests that increased levels of collagen are associated with an immunosuppressive tumor microenvironment. Experiments using 3D cultures with different collagen densities revealed that T cell proliferation and infiltration were significantly reduced in the high collagen density matrix [45]. Notably, recent studies have revealed bi-allelic mutations in AEBP1 in patients with Ehlers–Danlos syndrome (EDS), a heritable connective tissue disorder [46,47]. Genetic alterations of AEBP1 in EDS patients affect the discoidin domain of ACLP, which is required for interaction with collagen, and the disrupted ACLP function leads to defective collagen assembly in those patients [46]. A subsequent study demonstrated that ACLP enhances the stiffness, toughness, and tensile strength of collagen fibers [18]. Taken together with those studies, our results suggest that stromal ACLP may suppress intratumoral infiltration of CD8+ T cells by interacting with collagen in OSCC tissues.

There are several limitations in this study. First, the mechanism by which ACLP activates CAF is not fully understood. As ACLP is a secreted protein that associates with ECM, it may activate ECM-related signaling. A recent study showed that ACLP serves as a WNT ligand in hepatic stellate cells (HSCs) [48]. ACLP activates HSCs by inhibiting PPARγ through activation of WNT signaling, which leads to exacerbation of nonalcoholic steatohepatitis (NASH) [48]. Li et al. showed that ACLP also activates CAFs by repressing PPARγ via WNT signaling in pancreatic cancer, which suggests ACLP may activate CAFs via a similar mechanism in OSCC [39]. Second, we observed that ACLP not only activates CAFs; it also promotes migration, invasion, and chemoresistance in OSCC cells. However, the mechanism by which ACLP activates cancer cells remains unknown, and specific receptors that interact with ACLP in cancer cells have not been identified. Further studies are needed to clarify the molecular function of ACLP in malignant diseases.

## 5. Conclusions

In summary, we showed that elevated stromal ACLP expression is associated with the progression of OSCCs. Upregulation of ACLP leads to activation of CAFs, and CAF-derived ACLP promotes cancer cell migration, invasion, and in vivo tumorigenesis. We also found that high ACLP expression may contribute to the formation of an immunosuppressive tumor microenvironment by interacting with collagen. These findings suggest ACLP is a potential therapeutic target for treating OSCCs.

## Figures and Tables

**Figure 1 cancers-15-04303-f001:**
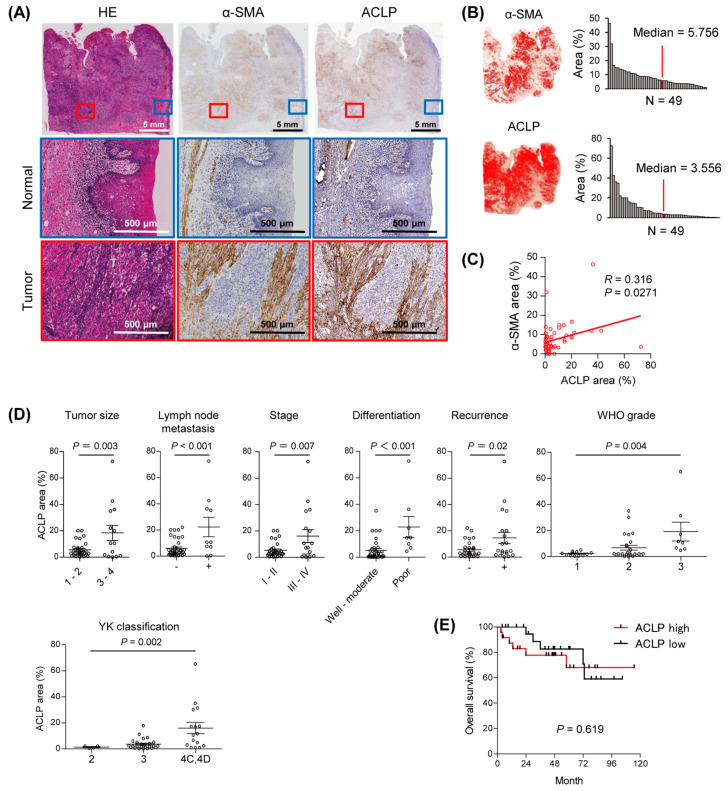
Stromal expression of ACLP in primary OSCC. (**A**) Immunohistochemical staining for α-SMA and ALCP in a representative OSCC tissue sample. Magnified views of normal and tumor areas are indicated by boxes and shown below. (**B**) Areas positive for α-SMA and ALCP in the representative sample are shown on the left. Summarized results for α-SMA- and ALCP-positive areas in OSCC tissues (*n* = 49) are shown on the right. (**C**) Correlation between the α-SMA- and ALCP-positive areas. (**D**) Correlations between the extent of ACLP-positive areas and clinicopathological characteristics in primary OSCCs. (**E**) Kaplan–Meier curves showing the effect of ACLP expression (high, ≥3.556%; low, <3.556%) on overall survival of OSCC patients.

**Figure 2 cancers-15-04303-f002:**
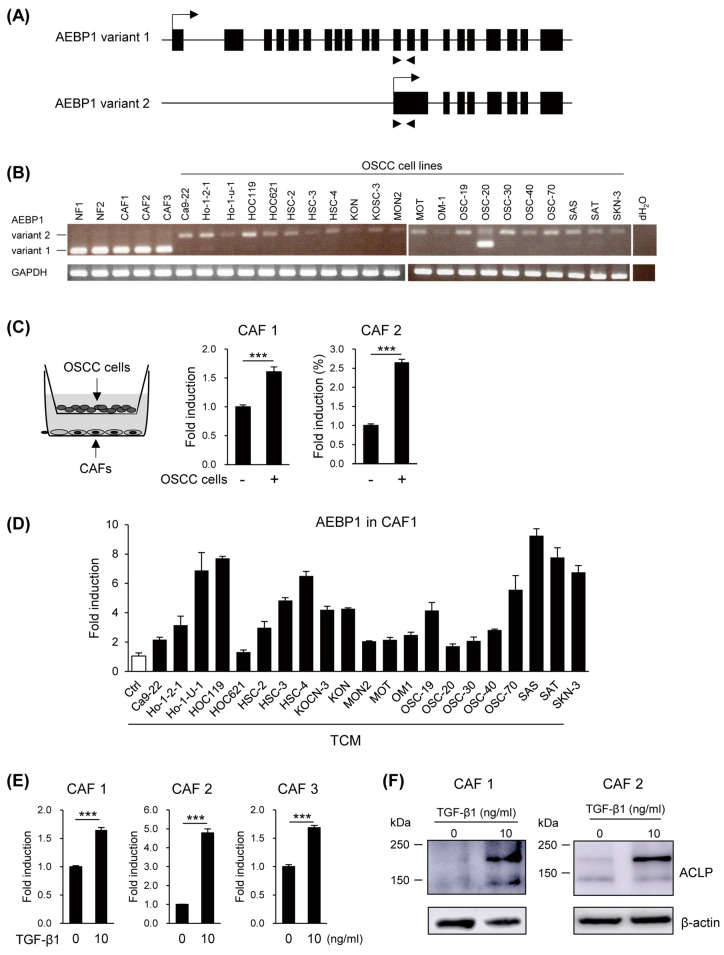
Expression of *AEBP1*/ACLP in CAFs. (**A**) Structures of the *AEBP1* gene. RT-PCR primers used to detect transcriptional variants are shown by arrows below. (**B**) RT-PCR analysis of *AEBP1* variants in normal fibroblasts (NF), CAFs, and OSCC cell lines. (**C**) Indirect co-culture of CAFs and OSCC cells (SAS) upregulates *AEBP1* in CAFs. A schema of the co-culture is shown on the left. Results of qRT-PCR of *AEBP1* in the indicated CAFs are shown on the right (*n* = 3). Error bars represent SEMs. (**D**) qRT-PCR of *AEBP1* in CAFs treated with TCMs derived from the indicated OSCC cell lines (*n* = 3). Error bars represent SEMs. (**E**) qRT-PCR analysis of *AEBP1* in the indicated CAFs treated with TGF-β1 (*n* = 3). Error bars represent SEMs. (**F**) Western blot analysis of ACLP in CAFs treated with TGF-β1. *** *p* < 0.001. The uncropped bolts are shown in Appendix A.

**Figure 3 cancers-15-04303-f003:**
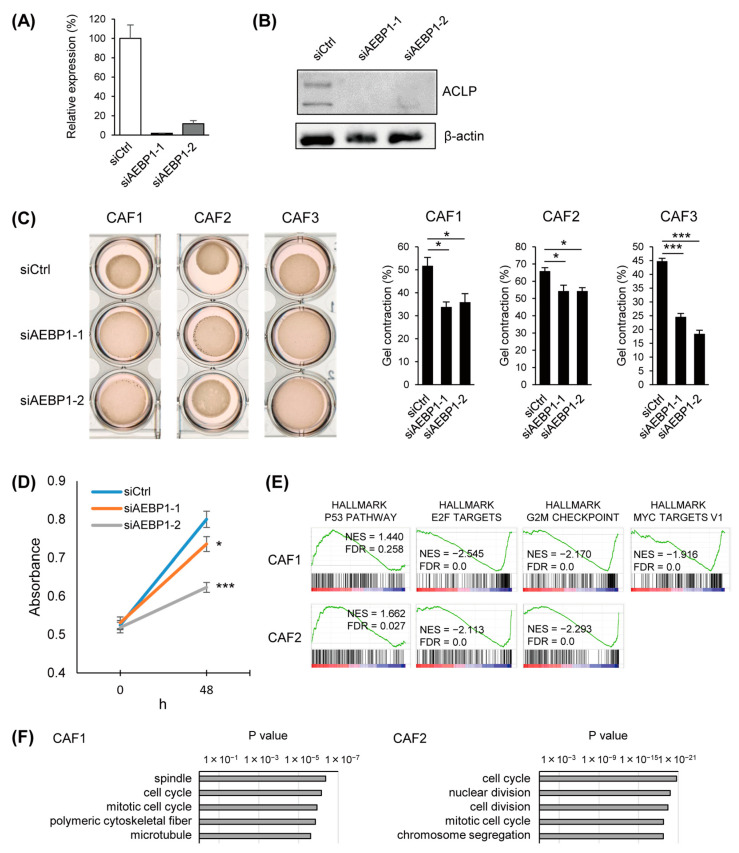
Functional analysis of *AEBP1*/ACLP in CAFs. (**A**) qRT-PCR of *AEBP1* in CAFs transfected with indicated siRNAs (*n* = 3). Error bars represent SEMs. (**B**) Western blot analysis of ACLP in CAFs with indicated siRNAs. The uncropped bolts are shown in Appendix A. (**C**) Collagen gel contraction assays in the indicated CAFs transfected with indicated siRNAs. Representative results are shown on the left; summarized results are on the right (*n* = 3). Error bars represent SEMs. (**D**) Results of cell viability assays with CAFs (CAF2) transfected with indicated siRNAs (*n* = 6). Error bars represent SEMs. (**E**) GSEA of genes in the indicated gene sets using the microarray data obtained from the indicated CAFs with *AEBP1*/ACLP knockdown. (**F**) Gene ontology analysis of genes downregulated (>1.5-fold) by *AEBP1*/ACLP knockdown. * *p* < 0.05, *** *p* < 0.001.

**Figure 4 cancers-15-04303-f004:**
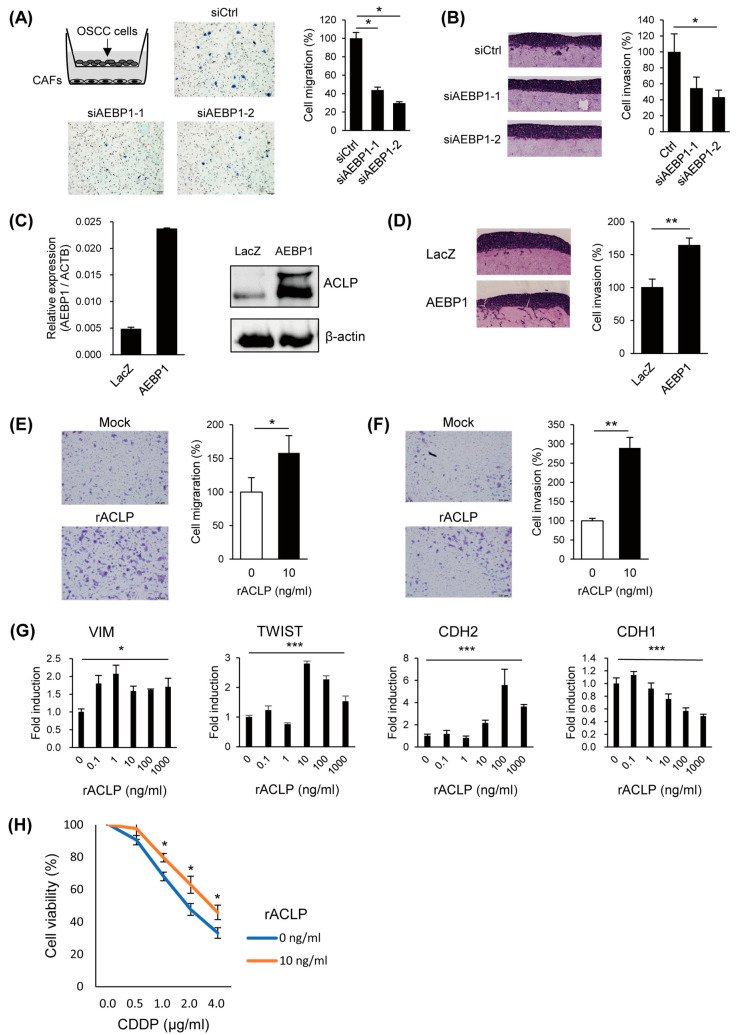
Functional analysis of CAF-derived *AEBP1*/ACLP on OSCC cells. (**A**) Migration assays using OSCC cells (SAS) and CAFs (CAF1). A schema of the assay is shown on the upper left. Representative results using CAFs transfected with indicated siRNAs are shown on the left. Summarized results are shown on the right (*n* = 3). Error bars represent SEMs. (**B**) Collagen gel invasion assays using SAS cells and CAFs. Representative results using CAFs transfected with indicated siRNAs are shown on the left; summarized results are on the right (*n* = 3). Error bars represent SEMs. (**C**) qRT-PCR analysis of *AEBP1* (left) and Western blot analysis of ACLP (right) in CAFs infected with indicated lentiviral vectors (*n* = 3). Error bars represent SEMs. The uncropped bolts are shown in Appendix A. (**D**) Collagen gel invasion assays using SAS cells and CAFs infected with indicated lentiviral vectors. Representative results are shown on the left; summarized results are on the right (*n* = 3). Error bars represent SEMs. (**E**,**F**) Transwell migration (**E**) and invasion (**F**) assays using SAS cells treated with or without recombinant ACLP. Representative results are shown on the left; summarized results are on the right (*n* = 3). Error bars represent SEMs. (**G**) qRT-PCR of EMT markers in SAS cells treated for 24 h with the indicated concentrations of recombinant ACLP. (**H**) Cell viability assays using SAS cells treated with the indicated concentrations of rACLP and CDDP. * *p* < 0.05, ** *p* < 0.01, *** *p* < 0.001.

**Figure 5 cancers-15-04303-f005:**
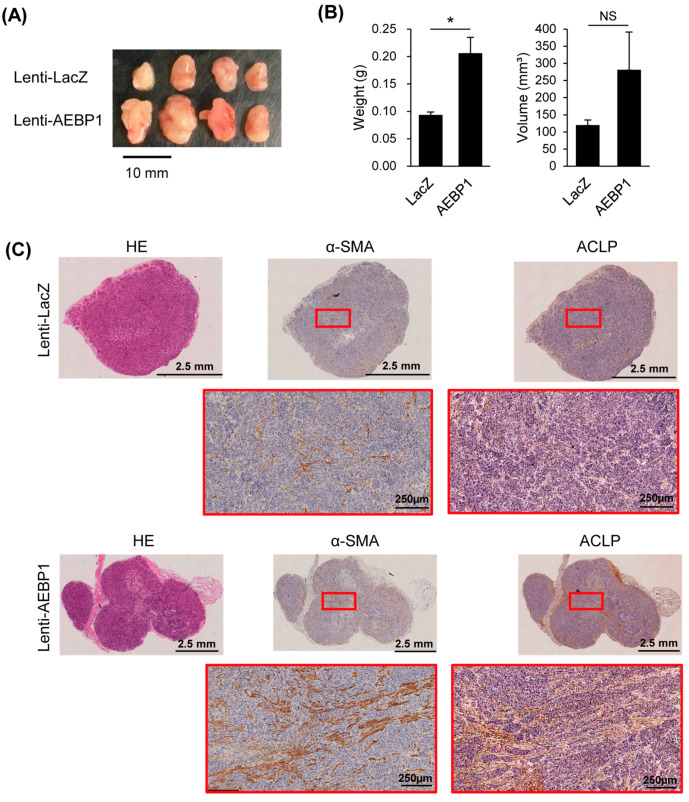
CAF-derived *AEBP1*/ACLP promotes in vivo tumorigenesis by OSCC cells (SAS) in a xenograft model. (**A**) Photographs of resected tumors with CAFs (CAF1) infected with indicated lentiviral vectors (*n* = 4). (**B**) Weights and volumes of the tumors (*n* = 4). (**C**) Immunohistochemical analysis of α-SMA and ACLP in xenograft tumors with CAFs infected with indicated lentiviral vectors. Magnified views of the boxed areas are shown below. * *p* < 0.05.

**Figure 6 cancers-15-04303-f006:**
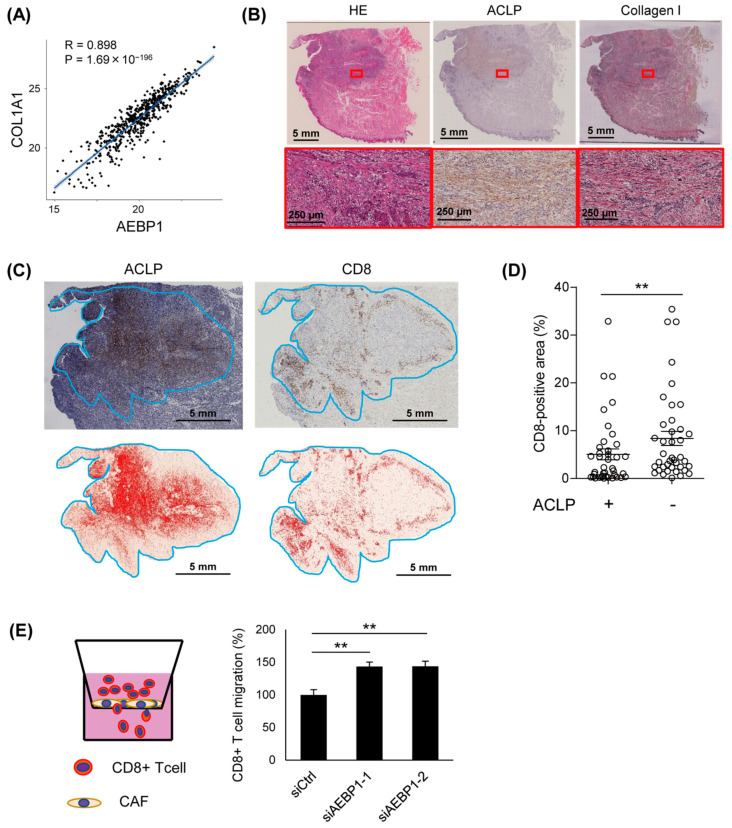
ALCP expression correlates positively with collagen expression and correlates negatively with intratumoral CD8+ T cell infiltration in primary tumors. (**A**) Correlation between mRNA expression of *AEBP1* and *COL1A1* in primary HNSCCs in TCGA dataset. (**B**) Immunohistochemical staining of ACLP and collagen I in a representative OSCC tissue sample. Magnified views of boxed areas are shown below. (**C**) Immunohistochemical staining of ACLP and CD8 in a representative OSCC tissue sample. Tumor areas, including invasive front regions, are indicated by blue lines. Areas positive for ALCP and CD8 are shown below. (**D**) Summaries of CD8-positive areas in ACLP-positive and -negative regions in OSCC tissues (*n* = 40). Error bars represent SEMs. (**E**) Transwell migration assays using CD8+ T cells. A schema of the assay is shown on the left. Summarized results using CAFs (CAF2) transfected with indicated siRNAs are shown on the right. ** *p* < 0.01.

## Data Availability

The Gene Expression Omnibus accession number for the microarray data is GSE234220. Additional data may be made available upon request.

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
