# Peer review of "ACLP Activates Cancer-Associated Fibroblasts and Inhibits CD8+ T-Cell Infiltration in Oral Squamous Cell Carcinoma"

_cancers, 2023, doi:10.3390/cancers15174303_

Round 1

Reviewer 1 Report

In the manuscript titled ACLP activates cancer-associated fibroblasts and inhibits CD8+ T-cell infiltration in oral squamous cell carcinoma,this study contains some interesting findings and are valuable for the understanding of the function of AEBP1 in CAFs. However, some revisions should be performed:

1 Materials and Methods2.1 the method of obtaining CD8 + T cells should has more detail

2 Figure2.D and G the picture is not very clear

3 Figure3 C,The conclusion that ACLP promotes CAF activation through collagen gel contraction is not accurate. Please clarify or change the expression

4 The first paragraph of the discussion, the fourth to last line, may be insufficiently well expressed, and the relationship between CAF activation and ACLP is not yet clear due to the lack of exploration of the pathway mechanism in this article

Author Response

We thank this reviewer for his/her very helpful comments to our manuscript.

Materials and Methods2.1 the method of obtaining CD8 + T cells should has more detail

Response: We added methods to obtain CD8+ T cells in the Materials and Methods section.

Figure2.D and G the picture is not very clear

Response: We agree that Figure 2D was not clear. Because the time until re-submission deadline is limited, we removed Figure 2D. We corrected brightness of Figure 2F (former Figure 2G).

Figure3 C, The conclusion that ACLP promotes CAF activation through collagen gel contraction is not accurate. Please clarify or change the expression.

Response: We corrected the sentence to “ACLP contributes to ECM remodeling by CAFs”.

The first paragraph of the discussion, the fourth to last line, may be insufficiently well expressed, and the relationship between CAF activation and ACLP is not yet clear due to the lack of exploration of the pathway mechanism in this article.

Response: We agree with the reviewer’s concern and added a following sentence in the first paragraph of Discussion. “However, further study is necessary to clarify the molecular function of ACLP in OSCC.”

Reviewer 2 Report

This manuscript studies ACLP in oral squamous cell carcinoma (OSCC), thus potentially benefiting future patients. Yet, some additions/changes detailed below are recommended.

As indicated by the authors in line 21, AEBP1 is also known as ACLP. Yet, in lines 69-72, ACLP is provided as an isoform of AEBP1. Accordingly, kindly make changes to ensure consistency and accuracy. Further, it is suggested to use the full names of AEBP1 and ACLP upon their first appearances.

Concerning line 101 of Section 2.1, a consent statement about the healthy donor is needed. Concerning lines 91-92, what does the “cell line” refer to? How were the CAFs isolated from the OSCC tissues? Concerning lines 98, 125, 158, 161, 170, and 185, what were the OSCC cells used? If they were primary cells from the OSCC tissues, how were they isolated? If they were established cell lines, what were they and where were they purchased/obtained from?

Concerning the Materials and Methods section, the current manuscript refers to Yorozu et al. 2020 for the detailed protocols of immunohistochemistry, RT-PCR, and western blot. Yet, Yorozu et al. 2020 itself does not provide those protocols. Instead, it refers to another paper. Also, it seems unnecessary to cite Yorozu et al. 2020 for the method of preparing tumor-conditioned medium since the method itself has been fully described in the current manuscript. Accordingly, clarification on the methods used is suggested in case our future readers would like to repeat any of the experiments.

Are the two panels at the bottom right corner of Figure 1 a part of Figure 1D? If so, kindly modify the layout to avoid any confusion. 

The ACLP bands in Figure 2D are too blurry to draw any conclusion. Is there any better WB result? Also, if possible, kindly assess the TGF-β concentrations in the TCM used in the experiment generating Figure 2E. Will neutralizing the TGF-β in the medium abolish any of AEBP1 induction effects shown? Further, according to Figure 2B, normal fibroblasts express AEBP1 variant 1 at a similar level compared to CAFs. Does this result go along with the conclusion drawn from Figures 1A and 1B stating, “expression of ACLP was higher in stromal cells within the tumor tissues than in adjacent normal tissues”? Will TCM and TGF-β also upregulate the AEBP1 variant 1 expression of a normal fibroblasts?

Concerning Figure 3D, there seems a significant difference in CAF proliferation between the siAEBP1-1 group and the siAEBP1-2 group. Kindly discuss the possible reasons behind it. Again, considering the siCtrl group in Figure 3D shows very faint ACLP band, it seems premature to conclude that the ACLP protein expression was eliminated by introducing siAEBP1-1 and siAEBP1-2. Also, Figure 3E is too blurry to read.

Concerning Figure 6E, does over-expressing ACLP further inhibit CD8+ T cell migration? Additionally or alternatively, will adding an antibody to neutralize ACLP abolish the inhibition of CD8+ T cell migration by ACLP?

The first “.” in line 241 seems unnecessary.

Assigning CDDP as the acronym of cisplatin is needed, for example in line 309.

Author Response

We thank this reviewer for his/her very helpful comments to our manuscript.

As indicated by the authors in line 21, AEBP1 is also known as ACLP. Yet, in lines 69-72, ACLP is provided as an isoform of AEBP1. Accordingly, kindly make changes to ensure consistency and accuracy. Further, it is suggested to use the full names of AEBP1 and ACLP upon their first appearances.

Response: As indicated by the reviewer, our description in the Simple Summary was incorrect. We removed “AEBP1” from the Simple Summary to avoid confusion. We also described full names of AEBP1 and ACLP upon their first appearance.

Concerning line 101 of Section 2.1, a consent statement about the healthy donor is needed.

Response: We added a consent statement about the heathy donor.

Concerning lines 91-92, what does the “cell line” refer to? How were the CAFs isolated from the OSCC tissues? Concerning lines 98, 125, 158, 161, 170, and185, what were the OSCC cells used? If they were primary cells from the OSCC tissues, how were they isolated? If they were established cell lines, what were they and where were they purchased/obtained from?

Response: OSCC cell lines were established by Japanese researchers and we obtained them from the Japanese Collection of Research Bioresources. We revised the manuscript and added references. Cell lines used in respective experiments are described in figure legends. We also added methods to isolate CAFs in the Materials and Methods section.

Concerning the Materials and Methods section, the current manuscript refers to Yorozu et al. 2020 for the detailed protocols of immunohistochemistry, RT-PCR, and western blot. Yet, Yorozu et al. 2020 itself does not provide those protocols. Instead, it refers to another paper. Also, it seems unnecessary to cite Yorozu et al. 2020 for the method of preparing tumor-conditioned medium since the method itself has been fully described in the current manuscript. Accordingly, clarification on the methods used is suggested in case our future readers would like to repeat any of the experiments.

Response: As suggested by the reviewer, we corrected the references for immunohistochemistry, RT-PCR and western blot analysis.

Are the two panels at the bottom right corner of Figure 1 a part of Figure 1D? If so, kindly modify the layout to avoid any confusion.

Response: As suggested by the reviewer, we corrected the layout of Figure 1.

The ACLP bands in Figure 2D are too blurry to draw any conclusion. Is there any better WB result?

Response: Because the time until re-submission deadline is limited, we could not re-perform the western blot analysis. So, we removed Figure 2D, and revised the manuscript accordingly.

Also, if possible, kindly assess the TGF-β concentrations in the TCM used in the experiment generating Figure 2E. Will neutralizing the TGF-β in the medium abolish any of AEBP1 induction effects shown?

Response: We agree with the reviewer’s comment. Because time until re-submission deadline is limited, we would like to address this issue in our future study.

Further, according to Figure 2B, normal fibroblasts express AEBP1 variant 1 at a similar level compared to CAFs. Does this result go along with the conclusion drawn from Figures 1A and1B stating, “expression of ACLP was higher in stromal cells within the tumor tissues than in adjacent normal tissues”?

Response: We agree with the reviewer’s concern. We consider that when normal and fibroblasts were cultured in the same medium, they express similar levels of AEBP1. However, we observed that levels of AEBP1 was robustly upregulated when they were co-cultured with OSCC cell lines, as shown in Figure 2E. We believe that this is consistent with our observation in clinical samples. A recent study by Li et al. (Cancer Lett, 544:215802, 2022: reference 36) showed higher levels of AEBP1 (ACLP) expression in CAFs than in normal fibroblasts by qRT-PCR. The discrepancy between our data and those by Li et al. may be caused by the difference in the cell culture condition. We consider that further study is necessary to clarify this point.

Will TCM and TGF-β also upregulate the AEBP1 variant 1 expression of a normal fibroblasts?

Response: We agree that this is a very important point. Because the time until re-submission deadline is limited, we would like to address this issue in our future study.

Concerning Figure 3D, there seems a significant difference in CAF proliferation between the siAEBP1-1 group and thesiAEBP1-2 group. Kindly discuss the possible reasons behind it.

Response: As shown in Figures 3E and 3F, our microarray analysis suggested that AEBP1/ACLP may promote CAF proliferation through modulating the cell cycle. We added following sentences in the first paragraph of the Discussion. “Our data also suggested that ACLP may promote proliferation of CAFs through modulating the cell cycle. However, further study is necessary to clarify the molecular function of ACLP in OSCC.”

Again, considering the siCtrl group in Figure 3D shows very faint ACLP band, it seems premature to conclude that the ACLP protein expression was eliminated by introducing siAEBP1-1 andsiAEBP1-2. Also, Figure 3E is too blurry to read.

Response: We corrected Figure 3D. We removed unnecessary parts from Figure 3E and enlarged the images.

Concerning Figure 6E, does over-expressing ACLP further inhibit CD8+ T cell migration? Additionally or alternatively, will adding an antibody to neutralize ACLP abolish the inhibition of CD8+ T cell migration by ACLP?

Response: We agree that this is a very important point, but we did not confirm whether overexpression of ACLP inhibits CD8+ T cell migration. In addition, as far as we know, there is no commercially available neutralizing antibody against ACLP.

Reviewer 3 Report

In this article, the author has discovered that cancer cells can mediate the expression of ACBL1 in CAFs. The expression of ACBL1 can activate fibroblasts, consequently promoting cancer cell migration, invasion, and the formation of tumors in vivo. This article is well-written, concise, and logically organized. Providing more detailed information could further facilitate the readers' understanding.

1. The author claimed that TGF originating from cancer cells can mediate ACBL expression in fibroblasts, yet the conditioned medium from cancer cell cultures contains a plethora of diverse cytokines. To demonstrate the predominant role of TGF, it is essential to compare the outcomes of cell treatment with conditioned medium before and after TGF removal by neutralizing or knocking down.

2. As a factor that promotes tumors and is highly expressed in tumors, why choose AEBP1 overexpression instead of knocking down or knocking out cell lines in tumorigenic experiments?

3. The author states the co-expression of AEBP1 and collagen can inhibits intratumoral infiltration by cytotoxic lymphocytes. Therefore, do tumors with AEBP1 overexpression exhibit lower immune cell infiltration in xenograft tumors?

Author Response

We thank this reviewer for his/her very helpful comments to our manuscript.

1. The author claimed that TGF originating from cancer cells can mediate ACBL expression in fibroblasts, yet the conditioned medium from cancer cell cultures contains a plethora of diverse cytokines. To demonstrate the predominant role of TGF, it is essential to compare the outcomes of cell treatment with conditioned medium before and after TGF removal by neutralizing or knocking down.

Response: We completely agree that it is a very important point. Because time until deadline of re-submission is limited, we would like to address this issue in our future study.

2. As a factor that promotes tumors and is highly expressed in tumors, why choose AEBP1 overexpression instead of knocking down or knocking out cell lines in tumorigenic experiments?

Response: For stable knockdown of AEBP1, we made a lentiviral shRNA vector. However, we did not success to establish CAFs with stable knockdown because the cells did not survive. It is unclear whether CAFs depended on AEBP1 for their survival or it was due to other technical problems, so we would like to address this issue in our future study.

3. The author states the co-expression of AEBP1 and collagen can inhibits intra tumoral infiltration by cytotoxic lymphocytes. Therefore, do tumors with AEBP1 overexpression exhibit lower immune cell infiltration in xenograft tumors?

Response: It is also a very important point. In the current study, we used nude mice in the xenograft experiments, so we could not evaluate immune cell infiltration. We would like to address this issue in our future study.

Reviewer 4 Report

Sekiguchi et al. have conducted comprehensive research to explore ACLP as a novel target in OSCC progression and invasion via interaction with fibroblasts and alteration of TiME. Using patient samples showing the elevation of ACLP in more advanced and malignant tumors in such a large cohort is valuable to the community to potentially use ACLP as a prognostic marker or even a therapeutic target. The in-vitro and inoculation work with manipulating ACLP expression helps validate the mechanism. However, the reviewer has a few concerns the authors should address before accepting this work.

Major

1)    Could the authors justify why only female nude mice were used for inoculation?

2)    Figure 1E: The reviewer would suggest performing segmentation analyses to separate metastatic diseases, stage, and recurrent disease, etc., within ACLP high and ACLP low groups, respectively. Because patients in these segments will have different overall survival, mixing them in the same group will mask the signal and make it hard to extract significant correlations.

3)    For experiments in Figures 3 and 4, N=3 is statistically unfavorable. Please repeat experiments to at least N=6.

4)    Figure 5: please indicate the number of samples or mice used in the experiment. The representative figure in 5A shows four tumors per group. Does that mean four mice were used for each group?

5)    Figure 5C: the representative figures of IHC for a-SMA and ACLP is not comparable. In the control group, the enlarged images (red box) show the center of a tumor. However, in the overexpression group, the images were taken at the interface of 2 or multiple tumor colonies. Naturally, more infiltrating immune cells and fibroblasts are found at the interface of tumor colonies than within the tumor. The authors used this as evidence to support that overexpression of ACLP promotes the expansion of tumor stroma, in this case, the control arm images should show the periphery of a tumor or the interface of multiple tumor colonies where it is more comparable to the figures shown in the overexpression arm. 

Minor

1)    Line 56: pembrolizumab is a better example of anti-PD-1 immunotherapy here since it is more widely used and approved in 1L therapy than nivolumab only in 2L.

2)    Figure 1A, right three panels: brightness is very low compared to the other two staining groups. Please adjust.

Author Response

We thank this reviewer for his/her very helpful comments to our manuscript.

Major

1) Could the authors justify why only female nude mice were used for inoculation?

Response: There are several reasons we used female nude mice. In general, male nude mice are more aggressive compared to females, which can make experimental manipulation difficult. We also tried to prevent injuries and damage within the cage caused by fights among male mice.

2) Figure 1E: The reviewer would suggest performing segmentation analyses to separate metastatic diseases, stage, and recurrent disease, etc., within ACLP high and ACLP low groups, respectively. Because patients in these segments will have different overall survival, mixing them in the same group will mask the signal and make it hard to extract significant correlations.

Response: As suggested by the reviewer, we performed statistical analysis by dividing the patients into ACLP-high and -low groups. We found that tumor size, differentiation and mode of invasion (YK classification) were significantly correlated with ACLP expression. We added the new data in Supplementary Table S3, and revised the manuscript accordingly.

3) For experiments in Figures 3 and 4, N=3 is statistically unfavorable. Please repeat experiments to at least N=6.

Response: We understand the reviewer’s concern, but we believe that differences were apparent in the all experimental data. Because the time until re-submission is limited, we do not have enough time to re-perform all the experiments.

4) Figure 5: please indicate the number of samples or mice used in the experiment. The representative figure in 5A shows four tumors per group. Does that mean four mice were used for each group?

Response: We are sorry that we did not specify the number of mice used in the experiment. We used four mice for each group. We added the number of samples in the figure legend.

5) Figure 5C: the representative figures of IHC for a-SMA and ACLP is not comparable. In the control group, the enlarged images (red box) show the center of a tumor. However, in the overexpression group, the images were taken at the interface of 2 or multiple tumor colonies. Naturally, more infiltrating immune cells and fibroblasts are found at the interface of tumor colonies than within the tumor. The authors used this as evidence to support that overexpression of ACLP promotes the expansion of tumor stroma, in this case, the control arm images should show the periphery of a tumor or the interface of multiple tumor colonies where it is more comparable to the figures shown in the over expression arm.

Response: In the current study, we used nude mice for the xenograft experiments, so that we could not evaluate immune cell infiltration in the xenograft tumors. Differences in the stromal cells were observed in the entire area of the tumors, not restricted to the periphery. We are sorry that the position of the red boxes in the ACLP overexpression tumor was not accurate, and magnifies views were from an intratumoral region. We corrected the position of the red box.

Minor

1) Line 56: pembrolizumab is a better example of anti-PD-1 immunotherapy here since it is more widely used and approved in 1L therapy than nivolumab only in 2L.

Response: As suggested by the reviewer, we replaced “nivolumab” with “pembrolizumab” and corrected the reference.

2) Figure 1A, right three panels: brightness is very low compared to the other two staining groups. Please adjust.

Response: As suggested by the reviewer, we adjusted the brightness.

Round 2

Reviewer 3 Report

The authors addressed the questions and the manuscript is now acceptable.